# Arthropod-Borne Viruses of Human and Animal Importance: Overwintering in Temperate Regions of Europe during an Era of Climate Change

**DOI:** 10.3390/microorganisms12071307

**Published:** 2024-06-27

**Authors:** Karen L. Mansfield, Mirjam Schilling, Christopher Sanders, Maya Holding, Nicholas Johnson

**Affiliations:** 1Vector Borne Diseases, Virology Department, Animal and Plant Health Agency, Woodham Lane, Addlestone, Surrey KT15 3NB, UK; karen.mansfield@apha.gov.uk (K.L.M.); mirjam.schilling@apha.gov.uk (M.S.); 2The Pirbright Institute, Ash Road, Pirbright, Woking, Surrey GU24 0NF, UK; christopher.sanders@pirbright.ac.uk; 3Virology and Pathogenesis Group, UK Health Security Agency, Porton Down, Salisbury SP4 0JG, UK; maya.holding@ukhsa.gov.uk; 4Faculty of Health and Medical Sciences, University of Surrey, Guildford GU2 7XH, UK

**Keywords:** virus emergence, mosquitoes, midges, ticks, diapause, temperature

## Abstract

The past three decades have seen an increasing number of emerging arthropod-borne viruses in temperate regions This process is ongoing, driven by human activities such as inter-continental travel, combined with the parallel emergence of invasive arthropods and an underlying change in climate that can increase the risk of virus transmission and persistence. In addition, natural events such as bird migration can introduce viruses to new regions. Despite the apparent regularity of virus emergence, arthropod-borne viruses circulating in temperate regions face the challenge of the late autumn and winter months where the arthropod vector is inactive. Viruses therefore need mechanisms to overwinter or they will fail to establish in temperate zones. Prolonged survival of arthropod-borne viruses within the environment, outside of both vertebrate host and arthropod vector, is not thought to occur and therefore is unlikely to contribute to overwintering in temperate zones. One potential mechanism is continued infection of a vertebrate host. However, infection is generally acute, with the host either dying or producing an effective immune response that rapidly clears the virus. There are few exceptions to this, although prolonged infection associated with orbiviruses such as bluetongue virus occurs in certain mammals, and viraemic vertebrate hosts therefore can, in certain circumstances, provide a route for long-term viral persistence in the absence of active vectors. Alternatively, a virus can persist in the arthropod vector as a mechanism for overwintering. However, this is entirely dependent on the ecology of the vector itself and can be influenced by changes in the climate during the winter months. This review considers the mechanisms for virus overwintering in several key arthropod vectors in temperate areas. We also consider how this will be influenced in a warming climate.

## 1. Introduction

Seasonality in temperate zones, those regions between 23.5° and 65.5° latitude north and south of the equator (Figure 1), is marked by dramatic changes to average temperature and day length that heavily influence arthropod ecology and, by extension, their ability to act as disease vectors [1]. The transition through the solar year (spring, summer, autumn and winter), each with different climatic conditions, is determined by a range of factors including latitude, the tilt of the Earth that varies the distance from the sun at different periods of the year, altitude and, for some regions, ocean currents [2]. Overlaying this are the impacts of global climate change, which are modifying long-term weather and climate patterns [3] that drive many aspects of arthropod vector behaviour.

Conditions in tropical regions, those around the equator, are generally conducive to year-round arthropod activity and continuous opportunities for virus transmission. However, even in these regions, seasonal variation in rainfall can lead to prolonged drought that can interrupt virus transmission. The climatic variation in temperate regions means that extreme diurnal temperatures can vary by over 20 °C. Especially during the late autumn, winter and early spring periods, temperatures are often below 0 °C for prolonged periods of time. As a result, there is often reduced arthropod activity during these periods. For arthropod-borne viruses to persist within temperate regions, mechanisms are required to overcome periods of the year when there is little or no vector activity [4,5]. Native arthropod species do successfully survive the winter period and emerge in spring, indicating that they may provide a route for virus overwintering.

A number of arthropod-borne viruses are established in temperate regions of Europe with the ability to successfully overwinter, and are detailed in Table 1.

A key mechanism that many arthropod species use to survive long periods of inhospitable conditions is diapause. This is a hormonally induced metabolic process which is triggered by environmental cues and results in a period of suspended development (dormancy). Not all arthropods have this survival mechanism. Adults of the mosquito *Aedes aegypti*, for example, generally do not undergo diapause, which restricts the species to tropical and sub-tropical regions where they remain active all year round. However, in this species, egg diapause can occur, facilitating its expansion into colder regions [6]. The related species *Aedes albopictus* undergoes diapause as embryonic larvae [7], and this has enabled its geographical expansion between continents [8]. In Europe, outbreaks of certain viruses such as dengue virus and chikungunya virus are vectored by *Ae. albopictus*, and therefore, this species may be one driver of viral emergence in temperate zones. However, such exotic arbovirus outbreaks tend to occur during the summer months, are geographically restricted, and do not appear to persist into the following year [9]. If a virus is to re-emerge in the spring months without further re-introduction, a mechanism is needed in which it persists. Three potential mechanisms for persistence in temperate regions are possible (see Figure 2 below). Firstly, the virus could persist in an arthropod vector, taking advantage of natural diapause mechanisms that enable the arthropod to survive extended cold or dry periods [10]. Secondly, the virus could persist in a vertebrate host that remains within the temperate region. Finally, the virus might persist in the environment. However, there is limited shedding of many arthropod-borne viruses from the vertebrate host and no evidence for environmental persistence outside of the host as is observed for some viruses such as foot-and-mouth virus [11]. 

This review focuses on the potential mechanisms of virus overwintering associated with the arthropod vector or the vertebrate host. Determining how arthropod-borne viruses achieve this is critical to predicting the timing and location of disease re-emergence events in temperate regions of northern Europe through overwintering. A further consideration is the impact of climate change on the winter periods in temperate regions. Increases in average winter temperatures will promote arthropod survival and extend the period over which vectors are active. This has become particularly critical in northern Europe with the emergence and spread of viruses such as West Nile virus [12] and most recently bluetongue virus serotype 3 during the late summer of 2023 in the Netherlands [13]. The continued spread of this virus during 2023 and its potential emergence in 2024 provide clear examples of the challenge virus overwintering in Europe poses for animal and public health.

## 2. Midge-Borne Virus Overwintering in Europe

In Europe, the most significant disease transmitted by midges within the genus *Culicoides* is bluetongue (BT). It is a non-contagious disease of ruminants, caused by bluetongue virus (BTV; family: *Reoviridae*, genus: *Orbivirus*). BTV infects all ruminants, with clinical signs ranging from mild to severe disease, although sheep tend to be the most severely affected. Cattle and other ruminants are not typically affected clinically, but may exhibit viremia and act as effective reservoirs [14]. BTV is transmitted between its ruminant hosts by the bites of *Culicoides* midges. BTV occurs in at least 29 serotypes, with varying pathogenicity and transmission characteristics between different strains. The distribution of BTV has changed significantly over the past three decades, with spectacular emergence into naïve temperate regions and an increased diversity of viral serotypes and strains in endemic regions [15]. BTV was historically considered an exotic pathogen of the tropics and subtropical regions, with only sporadic incursions into southern Europe reported prior to 1998. However, the emergence of different BTV strains in southern Europe in the following years was thought to have been facilitated by the impact of global climate change on the distribution of *Culicoides imicola*, [16], with indications that Palearctic *Culicoides* could support BTV transmission [17]. In 2006, BTV-8 was identified in the Netherlands and rapidly spread to neighbouring countries, representing a huge and unexpected expansion in the distribution of BTV. Perhaps counter to expectations, given the absence of adult midge vectors in the cold temperate winter, BTV-8 successfully overwintered in Europe to emerge in 2007. The following outbreak was the largest and most economically damaging outbreak of BTV in European history [18]. BTV was reported in countries as far north as the UK and Scandinavia [19]. In recent years, six serotypes of BTV (1, 2, 3, 4, 8, 16) have been present in Europe and are an ongoing threat to ruminant health.

Transmission of BTV in temperate regions is highly seasonal, with an apparent break in transmission through the winter and spring [20]. However, the mechanisms by which BTV is able to overwinter in Europe in the absence of vector activity and emerge in the following year are unclear. Suggested overwintering mechanisms of BTV include overwintering in long-lived insect hosts, extended duration of viremia in the ruminant host and vertical transmission in ruminants and vectors [21]. 

In contrast to many mosquito species, *Culicoides* are thought to overwinter as larvae [22]. In Europe, adult *Culicoides* are absent, or present at a small fraction of the summer abundance, for much of the winter. In the UK, for example, this period of absence typically lasts from December to March/April inclusive [23,24,25]. Adult emergence has been found to be negligible during this period [26]. The ‘vector free period’ where adults are absent in the UK is approximately 100 days, thought to be longer than adult survival. Adult *Culicoides* are absent from collections after the first or second ground frost of the winter [27]. A low vector competence of *Culicoides* species across Europe for BTV is typical, with less than 10% of individuals exposed to a viraemic blood meal supporting BTV infection and transmission, even for strains exhibiting transboundary spread [28]. Therefore, a large abundance of *Culicoides* is required for efficient BTV transmission to ruminants. Declaration of a Seasonal Vector Free Period (SVFP), where <5 *Culicoides* are collected in surveillance traps, is used to determine periods during outbreaks when the risk of transmission to ruminants is low and animal movement may recommence. It has been suggested that climate change will facilitate bridging of the SVFP, and therefore, overwintering *Culicoides* may be present for longer and emerge earlier. Indeed, changes in the phenology of *Culicoides* over the past forty years have been observed at one site in the UK, where the change of an increase in the length of the seasonal activity period would decrease the defined SVFP by 40 days. However, the response of populations to annual variation in climate was found to be dependent on local drivers such as environmental temperature and precipitation, and the period during which vectors are absent would still be sufficient to break transmission [29].

There is limited evidence for the vertical transmission of BTV from one generation of *Culicoides* to the next. Although detection of BTV RNA has been reported from *Culicoides* larvae [30], it is thought whole BTV particles cannot enter the developing eggs, preventing transovarial transmission [31]. Infection of adult *Culicoides* with live BTV has not been observed in nulliparous and newly emerged females [20,32]. Therefore, the SVFP presents a significant break in the transmission cycle of BTV. 

In temperate California, *Culicoides* collected during the winter were found to be positive for BTV infection during a period where infection in sentinel cattle was not observed [20]. The maintenance of transmission was suggested to be through survival of long-lived female *Culicoides* that acquired BTV infection during the transmission season [20]. Winter temperatures in California are, however, typically higher than those recorded during the Northern European winter, where *Culicoides* are absent from January to March, and above minimum thermal activity limits for the Palearctic *Culicoides* [33]. A low level of BTV transmission throughout the winter in intensive, indoor cattle production has been reported in Germany [34], although this remains the only report of vector-borne transmission to ruminants during the European winter. Indoor biting activity of *Culicoides* in autumn suggests that individuals may survive cold temperatures as adults within the warmer microclimates afforded by animal housing, particularly within indoor rearing systems [35]. Therefore, endophily of *Culicoides* in the autumn may allow transmission to occur within animal holdings or protect infected surviving adults from cold temperatures [36,37]. Further evidence, including age determination of *Culicoides* collected in animal sheds in winter, is required to confirm the likelihood of this overwintering mechanism. 

BTV could also persist within its ruminant hosts, through chronic or latent infection, transplacental transmission or horizontally through direct contact or sexual transmission. BTV has long been associated with a long viremia in ruminants, particularly cattle, with BTV genome detection by PCR possible for up to 222 days post-infection (dpi) in experimental infections [38,39]. Virus isolation from experimentally infected animals, however, was only possible up to 49 dpi, and infection of *Culicoides* vectors possible only when virus isolation was successful and only to 21 dpi [38]. These data suggest that the viral genome detected during the long ‘plateau’ phase of detection is not from infectious BTV particles, or that the BTV particles are not available for infection of vectors, due to insufficient titre or interactions with the host immune system. Therefore, it appears unlikely that the duration of viremia in cattle is sufficient to encompass the winter period. Assessment of the potential contribution of persistent endophilic *Culicoides* and extended viremia found the likelihood of either mechanism alone or combined was insufficient to explain the overwintering of BTV in Germany in 2006 and 2007 [40,41]. Latent persistence of BTV in the skin was suggested as a potential mechanism for the successful recovery of virus from sheep two months after infection [42], although this result has not been replicated.

Transplacental transmission, from infected mother to offspring, has also been suggested as an overwintering mechanism for BTV, and has been demonstrated for some tissue culture-adapted, live vaccine strains of BTV [43]. During the BTV-8 outbreak in Northern Europe, several incidents of BTV-positive calves and lambs born to infected cows and ewes were reported in the field and following experimental infections [44,45,46,47]. This mechanism shows clear potential for female ruminants infected with BTV in the autumn to birth infectious offspring in spring. Transplacental transmission was thought to be capacity-limited to a few BTV strains, including BTV 8, and was used as potential evidence for a live vaccine origin of the strain. More recent studies have demonstrated transplacental transmission in other low-passage field strains [48,49] suggesting that this is a potential route for BTV overwintering. 

Other overwintering mechanisms have been proposed, including the maintenance of infection in wild ruminant reservoirs [50]. A duration of BTV viremia of over 112 days has been reported in red deer, similar to that observed in cattle [51]. Infection of wild ruminants may allow silent transmission, though BTV transmission to *Culicoides* from wild reservoirs remains to be quantified [52]. There is no obvious single mechanism that enables overwintering of *Culicoides*-borne viruses including BTV in temperate regions. Overwintering may rely on multiple mechanisms across viruses, locations, environmental conditions and vectors. The long-term trend for contraction in winter periods where temperatures are not permissive for midge activity is a concern, as it will reduce the barrier to BTV overwintering through a range of mechanisms.

## 3. Mosquito-Borne Virus Overwintering in Europe

Europe has experienced outbreaks of mosquito-borne diseases where humans are the most susceptible vertebrate host, including dengue virus [53] and chikungunya virus [54]. These pathogens are presumed to have been transmitted by the invasive mosquito *Ae. albopictus* and do not involve an animal reservoir. They have also tended to be geographically restricted, with outbreaks of short duration that were rapidly brought under control. Subsequent outbreaks of these viruses have required re-introduction from tropical or sub-tropical regions, and there is no evidence that they have established in continental Europe.

By contrast, there are other mosquito-borne viruses present in Europe that cause repeated late-summer outbreaks of disease. The most significant of these is West Nile virus (WNV; family *Flaviviridae*, genus *Orthoflavivirus*). The virus had been introduced into Europe on multiple occasions during the 20th century, most likely through annual bird migration from Africa, causing detectable outbreaks in humans and horses, which are both dead-end hosts for the virus [55]. During the early decades of the 21st century, the virus was successfully established in southern Europe, recently spread further north to Germany in 2018 [12], and was detected in the Netherlands in 2020 [56]. Overwintering of WNV has generally been assumed in Europe, where repeated emergence has occurred in the same location over consecutive years. This has been supported by genomic sequencing, which demonstrated a close genetic relationship between viruses detected in successive seasons [57,58]. Of the three options for virus persistence between successive years (Figure 2), overwintering in the mosquito vector is considered the most likely means of virus persistence. However, detection of WNV in overwintering mosquitoes is rare, despite attempts at mass trapping of mosquitoes during the winter months in areas where the virus has been detected in animal hosts. WNV lineage 2 has been detected in overwintering (February to April 2019) *Culex pipiens* in the Czech Republic [59]. Detection of WNV lineage 2 has also been reported in a single pool of *Cx. pipiens* (March 2021) sampled in Germany [60]. In both studies, viral nucleic acid was detected by sensitive RT-PCR, but virus was not isolated despite attempts to propagate in cell culture, suggesting that virus is present at very low levels in vector populations. Experimental data show that the virus, despite being non-detectable in mosquitoes during diapause, can reactivate once the infected mosquitoes return to higher temperatures and metabolically emerge from diapause [61,62]. Koenraadt et al. [63] demonstrated that European *Cx. pipiens* in diapause were equally competent vectors for WNV compared to non-diapausing mosquitoes. Consequently, diapausing mosquitoes are likely contributing to viral persistence in temperate areas. Mathematical modelling that is taking the impact of rising temperatures in the UK into account predicts an increased risk of WNV outbreaks under prolonged periods of mosquito activity [64]. As a consequence, this might increase the number of WNV-positive mosquitoes entering diapause each autumn in temperate areas, which will most likely increase the chance for the virus to overwinter. However, increased temperatures during the summer months could also lead to increased larval mortality, which would then in turn reduce the number of adult females going into diapause [65]. 

A second virus causing repeated outbreaks of disease in Europe is Sindbis virus (SINV; family *Togaviridae*, genus *Alphavirus*), which was first detected in Sweden in 1967. Infection in humans causes a febrile illness characterised by a skin rash and joint inflammation that can persist long after the initial infection [66]. Phylogenetic analysis has suggested that the emergence of SINV in Europe was facilitated by a single introduction from sub-Saharan Africa with resultant spread to other Scandinavian countries [67]. Subsequently, the virus was detected in Germany in 2009, circulating in a sylvatic cycle [68] that has persisted and expanded into central Europe, although this does not appear to have resulted in human infections. An isolated study has detected SINV in overwintering *Cx. pipiens* from Sweden [69]. Again, virus genome was detected by RT-PCR, but at a low copy number, and the study was unable to isolate virus in cell culture.

A third emerging mosquito-borne arbovirus in Europe is Usutu virus (USUV; family *Flaviviridae*, genus *Orthoflavivirus*). Although closely related to WNV, infection in humans is usually asymptomatic, but highly virulent in certain bird species, including passerines and raptors [70]. The virus was first detected in African mosquitoes in the late 1950s and in Europe in the 1990s in Italy, and has become endemic in southern European countries [71]. USUV was first detected in Germany in 2011, with further spread of multiple lineages of the virus across Europe [72,73] resulting in bird die-offs in Germany [74], Belgium [75], the Netherlands [76] and potentially the United Kingdom [77]. There have been no reports of USUV detection in overwintering mosquitoes in Europe, but again, this may be due to the putative low prevalence of virus in overwintering female mosquitoes. However, transovarial transmission was observed when USUV was detected by RT-qPCR in an adult that emerged from a pool of larvae collected in Greater London in June 2023 (unpublished data). The timeline suggests that this transmission most likely occurred through a female that had emerged from diapause, indirectly indicating diapausing mosquitoes as a mechanism for the virus to overwinter.

The emergence and spread of these three viruses has occurred in an era of climate change that has caused warmer winters and wetter summers. Extreme weather events, such as flooding, are also more common. However, for viruses to persist between vector-active seasons, they still require a means of overwintering, and this is generally presumed to be in association with the vector, as viraemia in the vertebrate host, whilst high in certain bird species, is of short duration [78]. However, failure to detect virus in a range of mosquito species has led some authors to suggest that alternative mechanisms of overwintering should not be excluded [79]. Overwintering *Culex* mosquitoes rely on diapause, a physiological process that can be triggered at the larval stage and leads to physiological and behavioural changes in adult females [80]. Females that emerge in autumn are triggered to enter diapause through specific hormonal responses to a reduction in day length. This is referred to as the critical photoperiod (CPP). Other proposed stimuli for diapause in mosquitoes includes a reduction in temperature and nutrient availability, although these lack the consistency of CPP. Consequently, CPP is likely to be the primary driver for the onset of diapause in mosquitoes at higher latitudes. Environmental cues, such as CCP, prevent the release of juvenile hormone (JH) in emerging females [81], which would normally stimulate previtellogenic development of the ovaries and blood-seeking behaviour. In the absence of JH, the female does not seek a blood meal but searches for sugar sources, such as nectar, to build up fat reserves prior to entering diapause [82]. Other physiological responses include up-regulation of heat shock proteins to promote cold survival during adverse environmental conditions [83] and the triggering of a range of signalling pathways that lower metabolism to enable survival until conditions are more favourable [84]. However, these effects are reversible, and JH levels naturally increase during hibernation, stimulating ovary development and initiating host seeking behaviour when female mosquitoes become active [85]. This suggests that it takes several weeks to develop the full physiological response of adult diapause in *Cx. pipiens*, and the process is triggered relatively early in summer as daylight hours decrease. If a reduction in daylight hours is the critical stimulant for mosquito diapause, changes in climate will have little impact on its initiation. Peffers et al. [85] report that *Cx. pipiens* females enter diapause as the photoperiod reduces from 14 h to 13 h, with a critical photoperiod of 13.4 h of light/day leading to 50% diapause within a population (*Cx. pipiens* obtained from Michigan, USA, latitude 43 °N). Populations at latitudes higher than this would be expected to have CPP at earlier time-points in the year. For example, London (latitude 51.5 °N) experienced 13.4 h of daylight on 31 August 2022, suggesting the point where mosquitoes will begin physiological triggering of diapause and therefore, host-seeking behaviour should start to decline.

Several field surveys have reported evidence for the seasonal activity of *Culex* spp. in the UK. Field studies in southern England have found, through a combination of resting and active mosquito trapping, that *Cx. pipiens* and *Cx. modestus* were active from June onwards, with populations peaking during August and September before declining [86,87]. A more recent survey in a central London location showed that *Cx. pipiens* s.l. numbers peaked in August [88]. Combined, these findings suggest that CCP thresholds are important for understanding mosquito population dynamics behaviours, particularly host-seeking behaviour and mating, and by extension, population size and pathogen transmission risk. These datasets will be critical for monitoring the impact of climate change on mosquito activity and abundance, particularly the point in the year when overwintering mosquitoes begin host-seeking behaviour.

Critically for virus transmission, the stimulus for diapause leads female mosquitoes to favour sugar sources over blood, so a lower proportion of the mosquito population will actively host-seek during late summer and autumn. This in turn reduces the number of mosquitoes acquiring virus infection through feeding on viraemic hosts while simultaneously reducing onward transmission of a virus to naïve vertebrate hosts. That therefore leaves two further routes of mosquito infection that can facilitate overwintering of virus: vertical transmission from the previous generation to overwintering females, or venereal transmission from infected males to females during mating. However, this latter route would also require vertical transmission to males as a prerequisite, as they do not take blood meals. In North America, numerous studies have investigated persistence of West Nile virus [5,89,90,91]. Vertical transmission has been demonstrated for WNV in *Culex* spp. females from North America, in both natural and experimental paradigms [92]. Field studies investigating WNV overwintering in Californian *Culex* mosquitoes suggested that vertical transmission played a key role in persistence of virus from one season to the next [93]. A recent study in Sweden investigated the vertical transmission of SINV in *Culex* vectors, both in the field and experimentally [94]. Vertical transmission in *Cx. pipiens pipiens* and *Cx. torrentium* was confirmed in the field through detection of SINV RNA in field-collected egg rafts and emerging adults. In addition, experimentally infected *Cx. pipiens molestus* females produced adult offspring containing SINV RNA at emergence. Combined, these studies suggest that vertical transmission may be a mechanism by which SINV is able to persist throughout cold winters [94], but this has not been conclusively demonstrated for other virus–vector combinations in temperate areas. Climate change that increases winter temperatures and extends the period that mosquitoes are active will enhance mosquito survival but not the underlying mechanisms that enable viruses to overwinter with the vector.

## 4. Tick-Borne Virus Overwintering in Europe

In Europe, there are two tick-borne flaviviruses affecting human and livestock populations (Table 1). Tick-borne encephalitis virus (TBEV) and louping ill virus (LIV) share close genetic and antigenic homologies and are endemic in many regions of Europe. LIV is present in the British Isles, although in recent decades, it has also been reported in parts of Denmark and Norway [95,96,97]. TBEV has a much wider distribution across Europe and Asia, and is present across a number of northern European countries, including Norway, Denmark, Estonia, Finland, Norway, Sweden, Latvia, Lithuania and the United Kingdom [98,99]. There are several mechanisms that contribute to the survival of TBEV and LIV during winter in temperate areas, mainly thought to be through maintenance of virus in the tick vector. *Ixodes ricinus* is the main vector of LIV; and although evidence suggests at least 17 species of tick may be competent vectors of TBEV, *Ix. ricinus* and *Ix. persulcatus* are considered the main TBEV vectors and also the main tick species where TBEV RNA is detected [100,101]. Therefore, these tick species are most likely to contribute to viral overwintering in temperate areas. Temperature and humidity are the two key parameters that dictate tick activity. Ticks become active when temperatures are consistently above 7 °C [102]. However, as temperatures rise above 15 °C, the risk of desiccation increases, and questing ticks will return to the leaf litter layer to re-hydrate. Although *Ixodes* species are cold resistant, particularly species such as *Ix. uriae* [103], temperatures below −15 °C are lethal to most tick species found in temperate regions [104].

*Ixodes ricinus* and *Ix. persulcatus* take one blood meal per instar and are three-host ticks, feeding on a broad diversity of vertebrate hosts. Each of their life stages takes approximately 1 year to progress from one developmental stage to the next. Reaching the mature adult stage will therefore usually take 3 years, although this can range from 2 to 6 years [105]. Engorged, mated adult females (Figure 3) return to the leaf litter layer or a similar humid environment and produce eggs. The long lifespan and overwinter survival of these tick species facilitates the long-term maintenance of TBEV geographic foci and makes the vectors themselves a viable TBEV reservoir [105,106]. Primarily, the persistence of viruses in ticks is facilitated through transstadial transmission, where a pathogen acquired during one life stage is transferred through to the next life stage [107,108]. This is particularly important for transmission through infected larvae that overwinter and then feed as infected nymphs the following year. Interestingly, despite transstadial transmission being widely referenced as a key mechanism for the maintenance of TBEV and LIV, experimentally derived evidence indicated that the transmission rate between life stages, at <10%, may be lower than previously anticipated [108]. This may contribute to the low TBEV infection rates, often below 1%, that are detected in questing *I. ricinus*. 

Both *Ix. ricinus* and *Ix. persulcatus* are exophilic, exhibiting a seasonal questing behavior that varies by species, life-stage and latitude. Temperature, relative humidity, saturation deficit, rainfall and daylight hours are all important factors in seasonal activity. They tend to have low winter activity when temperatures are too low for questing or development to the next life stage. Diapause (described above) is a protective mechanism where ticks enter a state of dormancy, with a fixed latency period, entered ahead of seasonally unfavourable conditions [109]. This period ensures that tick questing activity aligns with optimal conditions for survival [110]. Quiescence, also a period of dormancy, is a state that can occur as an immediate response to unfavourable conditions. This may be entered if unfavourable conditions still exist at the end of a fixed period of diapause [109]. Despite being important for ticks and therefore virus survival, under experimental conditions, evidence suggests diapause may have a negative effect on TBEV titres in unfed nymphal ticks. In addition, virus titres may drop over time. Both of these factors may contribute to low virus prevalence in ticks [111,112]. TBEV has been shown to have a temperature-sensitive ‘riboswitch’, a sequence of RNA that alters genome secondary folding and could prevent translation at lower temperatures [113]. The breakpoint temperature, when the RNA may then unfold and replication commences, was found to vary depending on the climate from where the virus originated. Therefore, the impact of winter temperatures on TBEV may vary depending on the virus strain and climate where it is present [113]. Additionally, as shown in a study modelling the effects of a changing climate on the seasonality of Lyme disease, a bacterial vector-borne disease transmitted by *Ix. ricinus*, higher temperatures during the winter months would lead to an increase in the number of questing ticks in late winter and early spring [114]. Consequently, this would not only prolong the season of Lyme disease but also TBEV and increase the chance for the virus to overwinter.

Adult ticks are often reported as not important in the maintenance of TBEV, as their feeding preference is not the main small mammal reservoir hosts. However, transovarial transmission is another mechanism that may to some extent facilitate the maintenance of TBEV overwintering. Female ticks can survive winter diapause; therefore, even a small transovarial transmission rate to the next filial generation could potentially be important at maintaining the virus in an area. Laboratory experiments found transmission occurred from approximately 20% of infected females and from these, a filial transmission rate (the proportion of infected larvae in a clutch) of 0.23–0.75% [114]. It should be noted that many tick species produce in excess of 1000 eggs in a clutch. A later study found a lower rate of maternal transmission, with transovarial transmission occurring in just 3.3% of females [115]. Field experiments have recovered unfed larvae in the fur of small mammals, that were infected with TBEV [116], although this is not conclusive evidence of transovarial transmission. It has been argued that despite this low rate, given the large number of larvae that co-feed on small mammalian reservoir hosts, often in the absence of nymphs, transovarial transmission may be another route contributing to the maintenance of TBEV infection through winter [117,118]. In contrast to TBEV, transovarial transmission has not been reported for LIV [119].

Initial evidence indicates that TBEV reservoir hosts may also have a role in the maintenance of the virus over winter. A field study conducted in Finland found evidence suggesting the European and Siberian subtypes of TBEV may cause extended latent infections in rodents. It was found that several months after the tick feeding season, TBEV RNA was detectable in rodent brain and internal organs. Further research is required to establish this potential role of virus persistence in rodent hosts and its contribution to the overwintering of the virus [120]. It is also critical to find further evidence of virus in vertebrate tissues during the winter months to identify other potential reservoir hosts.

Whether vertebrate hosts contribute to overwinter survival of LIV is unclear; however, viraemia is usually reported to be transient. Sheep and red grouse are the main animal hosts of LIV, both producing post-infection viraemia sufficient for vector-host-vector transmission [121]. Both develop clinical disease that can be fatal to both species, with red grouse suffering particularly rapid mortality at high rates, found to be 79% under laboratory conditions [121]. Due to this rapid and high mortality, it is not thought that red grouse infect large numbers of ticks. However, the extent of the role of sheep in viral persistence may be reduced in areas of good sheep management through the use of acaricides and previously through vaccination [122]. 

There remains considerable uncertainty on the overwintering of tick-borne viruses within their reservoir hosts, in addition to ticks, and the relative contributions each mode of transmission makes. Further research is needed for both viruses, though particularly LIV, to further understand the mechanisms within the tick and possible involvement of animal hosts to ascertain their relative contribution to maintenance through winter. As winters become milder and shorter under the influence of climate change, the increased survival of ticks and potential viraemia in mammal reservoirs will increase the likelihood of virus overwintering and increasing the incidence of subsequent years. This effect is being observed at higher latitudes, as *Ix. ricinus* populations are being detected further north than previously reported [123], and cases of TBEV are occurring in new locations [124].

## 5. Conclusions

In order to establish persistence in temperate areas, emerging arthropod-borne viruses utilize mechanisms to maintain them through the winter months during periods of low temperatures and low vector activity. Yet, surprisingly little is known about these mechanisms. Despite very low temperatures experienced in northern latitudes, there is no evidence for prolonged environmental survival of arthropod-borne viruses. Where pathogens do persist between seasons, their survival is most likely facilitated through association with a host or vector. Temperate areas have vector-competent arthropod populations, as evidenced by the repeated emergence of pathogens from southern latitudes. However, the lower temperatures experienced in temperate regions alone will inhibit virus replication within an infected arthropod vector, and it is unclear what effect, if any, the physiological actions of diapause have on virus replication and persistence [10]. There are clear gaps in understanding the relative contributions that vertical and transstadial transmission make towards overwintering in arthropod vectors. The persistence and long-term establishment of such pathogens in higher latitudes is limited during the winter period that restricts vector activity. This is changing in response to an altering climate, most noticeably by the spread of ticks and tick-borne viruses to higher latitudes [125].

Identifying and characterizing persistence mechanisms in arthropod vectors would enable the design of effective vector control strategies. Specifically, predicting conditions that are adequate for the cessation of diapause could identify times of the year that would be appropriate for enhanced surveillance and vector control strategies by animal and public health agencies. This will detect disease emergence at the earliest possible time, prior to transmission to susceptible animals and humans. Moreover, understanding the vector competence of temperate arthropods, combined with their broader ecology and their contribution to pathogen overwintering success, would elucidate which diseases are likely to emerge in a given temperate region. Here, models that combine vector behaviour and environmental factors have a significant role to play in predicting arthropod-borne disease emergence and persistence [126]. As several infectious pathogens are not host-specific and may have cryptic transmission networks, there are likely to be a number of strategies available to facilitate overwintering, which when combined with a changing global climate, will increase the potential for tropic and sub-tropic pathogens to become established in temperate zones.

## Figures and Tables

**Figure 1 microorganisms-12-01307-f001:**
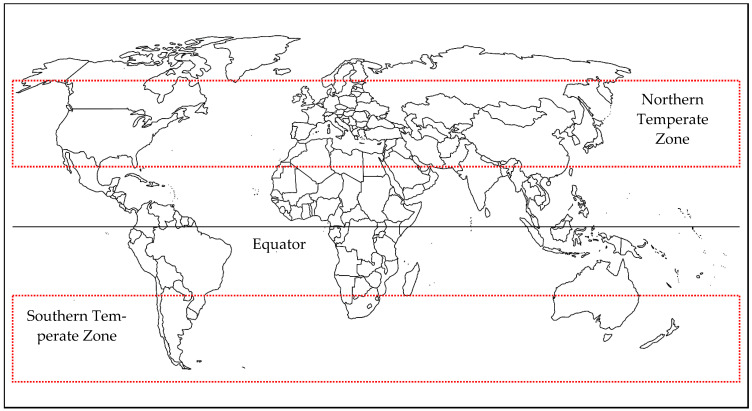
Northern and southern temperate zones to the north and south of the equator (red boxed areas). Within these zones, temperatures are persistently below that required for vector activity during the winter months and not permissive for arbovirus transmission.

**Figure 2 microorganisms-12-01307-f002:**
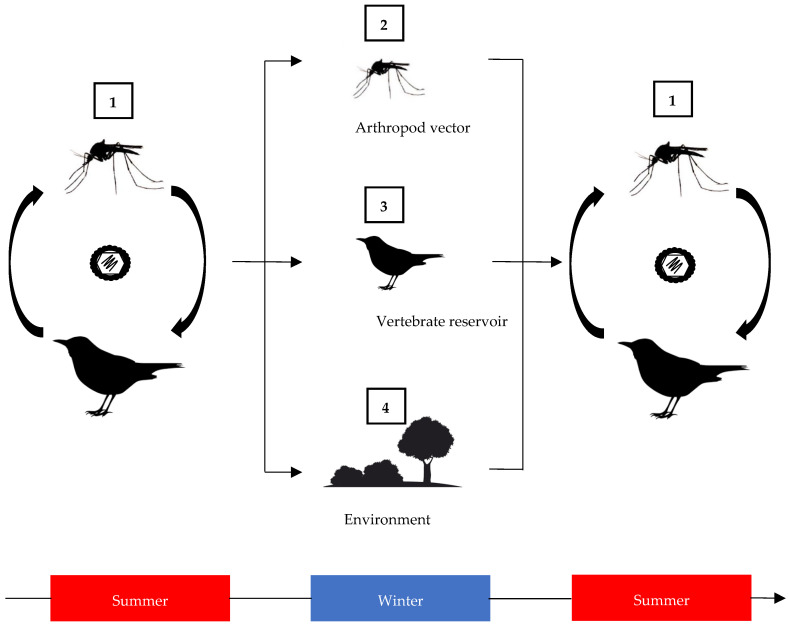
Schematic showing the potential pathways that would enable an arthropod-borne virus such as West Nile virus to overwinter in temperate zones. During permissive climatic conditions, virus exists in an epizootic transmission cycle between its vector and vertebrate host (1). During non-permissive periods of the year, such as the winter months, the virus could persist within the vector (2), the vertebrate reservoir (3) or potentially in the environment (4) until conditions change to restart the transmission cycle.

**Figure 3 microorganisms-12-01307-f003:**
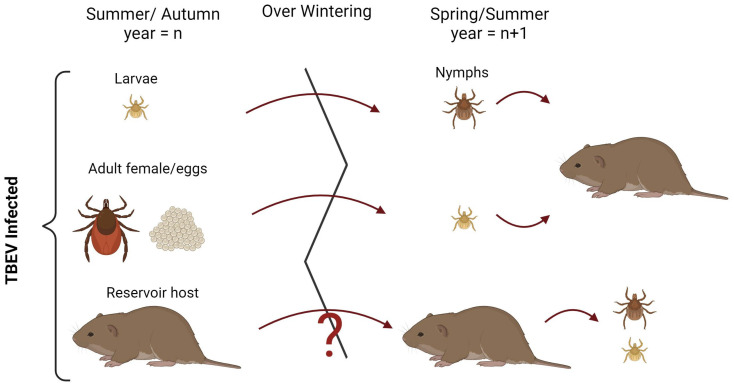
Potential routes for overwintering of tick-borne viruses. Virus could persist through transstadial transmission from one vector-active season (*n*) to the next (*n* + 1) within larvae that moult into the nymphal form (top). Alternatively, virus could be transmitted transovarially and infect the next season’s larvae (middle). There remains uncertainty over the ability of viruses to overwinter within a mammalian host and infect ticks during the following season (bottom). Created with BioRender.com.

**Table 1 microorganisms-12-01307-t001:** Arthropod-borne viruses transmitted by mosquitoes, midges, sandflies and ticks that successfully overwinter in Europe.

Virus Name	Family/Genus	Vector
Batai virus	*Peribunyaviridae*/*Orthobunyavirus*	Mosquitoes
Bluetongue virus	*Reoviridae*/*Orbivirus*	*Culicoides* spp.
Chatanga virus	*Peribunyaviridae*/*Orthobunyavirus*	*Aedes* spp.
Crimean Congo haemorrhagic fever virus	*Nairoviridae*/*Orthonairovirus*	*Hyalomma* spp.
Inkoo virus	*Peribunyaviridae*/*Orthobunyavirus*	*Aedes* spp.
Louping ill virus	*Flaviviridae*/*Orthoflavivirus*	*Ixodes ricinus*
Sindbis virus	*Togaviridae*/*Alphavirus*	*Culex* spp.
Schmallenburg virus	*Peribunyaviridae*/*Orthobunyavirus*	*Culicoides* spp.
Tahyna virus	*Peribunyaviridae*/*Orthobunyavirus*	Mosquitoes
Tick-borne encephalitis virus	*Flaviviridae*/*Orthoflavivirus*	*Ixodes* spp., *Dermacentor* spp.
Toscana virus	*Phenuiviridae*/*Phlebovirus*	*Phlebotomus* spp.
Usutu virus	*Flaviviridae*/*Orthoflavivirus*	*Culex* spp.
West Nile virus	*Flaviviridae*/*Orthoflavivirus*	*Culex* spp.

## Data Availability

No new data were created or analyzed in this study. Data sharing is not applicable to this article.

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
