# Peer review of "Arthropod-Borne Viruses of Human and Animal Importance: Overwintering in Temperate Regions of Europe during an Era of Climate Change"

_microorganisms, 2024, doi:10.3390/microorganisms12071307_

Round 1

Reviewer 1 Report

Comments and Suggestions for Authors

A well-structured and documented article. Everything perfect and very interesting. Congratulations!

Author Response

A well-structured and documented article. Everything perfect and very interesting. Congratulations!

Comment > Thank you for these very positive comments.

Reviewer 2 Report

Comments and Suggestions for Authors

The manuscript with the title "Arthropod-borne virus overwintering in temperate regions of Europe during an era of climate change" describes the effects of climate change on the capacity for several viruses of human and veterinary important viruses to overwinter in temperate European climates. The manuscript is very well written and I believe fills an important gap in the current review space of recent literature on the subject. The only "major" change that I would suggest is slightly changing the title to reflect the fact that the review focusses on viruses of human and veterinary importance. Other than that a few very minor, specific suggested edits as follows:

Line 237: Delete “furthermore”

Line 311. Check the term “nigureectar” and correct. 

Line 448: Delete “furthermore”

Author Response

The manuscript with the title "Arthropod-borne virus overwintering in temperate regions of Europe during an era of climate change" describes the effects of climate change on the capacity for several viruses of human and veterinary important viruses to overwinter in temperate European climates. The manuscript is very well written and I believe fills an important gap in the current review space of recent literature on the subject. The only "major" change that I would suggest is slightly changing the title to reflect the fact that the review focusses on viruses of human and veterinary importance. Other than that a few very minor, specific suggested edits as follows:

Comment > Thank you for these comments. We have amended the title to: “Arthropod-borne viruses of human and animal importance: Overwintering in temperate regions of Europe during an era of climate change.

Line 237: Delete “furthermore”

Comment > “furthermore” has been deleted from the manuscript

Line 311. Check the term “nigureectar” and correct. 

Comment > The word should be “nectar”, this has been corrected in the manuscript.

Line 448: Delete “furthermore”.

Comment > “furthermore” has been deleted from the manuscript

Reviewer 3 Report

Comments and Suggestions for Authors

The review presents a comprehensive overview of impact of climate change on emergence of arthropod borne infectious diseases. The authors have presented a detailed summarization of arthropod-borne viruses transmitted by mosquitoes, midges, sandflies and ticks that successfully overwinter & described potential pathways in permissive and non-permissive climate conditions. Understanding vector competence, adaptation and environment impact contributes for designing effective vector control strategies. Therefore, the review has great scientific interest and is suitable for publishing for the journal. However, there are few minor comments that need to be addressed.1-    I would suggest simplifying the rationale, broad impact and conclusion of the review for better understanding.

2-    The presentation and quality of figures can be improved, for example legend for figure-3 should be included for better understanding.

Author Response

The review presents a comprehensive overview of impact of climate change on emergence of arthropod borne infectious diseases. The authors have presented a detailed summarization of arthropod-borne viruses transmitted by mosquitoes, midges, sandflies and ticks that successfully overwinter & described potential pathways in permissive and non-permissive climate conditions. Understanding vector competence, adaptation and environment impact contributes for designing effective vector control strategies. Therefore, the review has great scientific interest and is suitable for publishing for the journal. However, there are few minor comments that need to be addressed.

1-    I would suggest simplifying the rationale, broad impact and conclusion of the review for better understanding.

Comment > Thank your for these comments. The rationale and scope of the review has been simplified within the abstract to “This review considers the mechanisms for virus overwintering in several key arthropod vectors in temperate areas. We also consider how this will be influenced in a warming climate.” Minor changes have been made to the Conclusions section.

2-    The presentation and quality of figures can be improved, for example legend for figure-3 should be included for better understanding.

Comment > Figure 3 and its legend have been revised to “Potential routes for overwintering of tick-borne viruses. Virus could persist through transstadial transmission from one vector-active season (n) to the next (n+1) within larvae that moult into the nymphal form (top). Alternatively, virus could be transmitted transovarially and infect the next seasons larvae (middle). There remains uncertainty over the ability of viruses to overwinter within a mammalian host and infect ticks during the following season (bottom). Created with BioRender.com.” The other figure legends have been amended to provide more clarity.